# Konjak: Live Visualization in Deep Neural Network Programming as a Learning Tool

Anonymous Authors *

Author's Affiliation

## ABSTRACT

Visualization in deep neural networks (DNNs) development could play a key role in helping novice programmers to inspect and understand a network structure. However, these visualizations are usually available only after the implementation of the DNN program. We propose combining a code editor with a live visualization of the DNN structure to assist machine learning novice users during the DNN code development. The user assigns operation blocks and input data size in the code editor, and the system continuously updates the network visualization. The visualization is also editable, where the user can directly use drag-and-drop operations to build a network. To our knowledge, we are the first to tightly combine text-based programming editor with live and editable visualization as an educational tool in DNN programming, which can help understand the concept of shape consistency. This paper describes the system's design rationale and presents an exploratory user study to evaluate its effectiveness as a learning environment.

**Index Terms:** Human-centered computing—Visualization—Visualization techniques—Treemaps; Human-centered computing—Visualization—Visualization design and evaluation methods

## 1 INTRODUCTION

In recent years, deep learning networks (DNNs) have been surging in many classical machine learning tasks, such as image classification [16, 18, 23, 30], object detection [15], text generation [11], and rendering [21]. As a sub-domain in machine learning, DNNs show impressive performance in most of the tasks above, even surpassing human-level accuracy. DNNs' powerful ability to dig insights from data makes it one of the most popular tools for researchers and practitioners to utilize in practice.

For non-expert machine learning users, like software engineers, medical doctors, and artists, DNNs are attractive for applications in their domains. Libraries like Tensorflow [1, 2] and Pytorch [29] provide high-level APIs to enable a more approachable model building without losing the flexibility needed by expert users to customize more details in their model. In general, DNNs are sequences of mathematical functions (a.k.a. layers) to process data in the form of multi-dimensional arrays (a.k.a. tensors), and arguments of these functions rule legal tensor shapes that can be processed. The programmer needs to consider the alignment between layer arguments and assumed input data shape at the early DNN programming stage. However, this is not an easy task for a novice machine learning user. We will detailedly describe the shape inconsistency error that the misalignment will lead to in Sect. 3.

By observing expert machine learning users and rethinking our own experience, we noticed that the network diagram plays a crucial role in DNN programming practice. DNN developers or researchers often draw node-link diagrams on a whiteboard for DNN structure communication [36] as well as scholarly communication. In the

---
*e-mail: Author's email

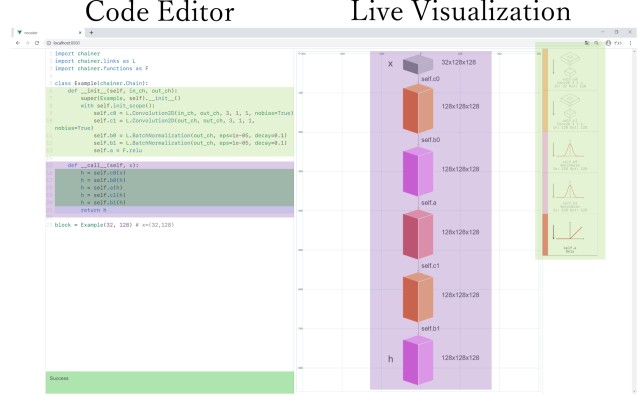

Code Editor     Live Visualization

Figure 1: Konjak shows live visualization next to the text-based code editor, providing continuous feedback on the Deep Neural Network structure to the programmers. The visualization also supports direct manipulation to edit the corresponding text code.

current DNN programming practice, visualization is optional and only available after the training phase. These visualization tools enable network validation at a very late stage in DNN modeling procedures, although visualization can guide the user during the code editing. Tools such as A Neural Network Playground [31] introduce manipulable visualization into DNN education to help explain the mechanism but do not show the corresponding text-based code. This prevents novice users from improving DNN programming skills, which usually involves text-based programming.

We propose Konjak, a system to augment a standard text-based code editor with an always-on and editable live network structure visualization to help machine learning novice programmers to learn DNN programming, as shown in Fig. 1. Konjak enables higher liveness than the current practice of DL system development, where the programmers can bidirectionally check and edit the DNN between synchronized code panel and visualization panel. We support an automated tensor shape checker to help users, especially novices, tackle a shape inconsistency error at where they occur in time. By comparing the codes and adjacent DNN visualization and repeating edit on either of them, the programmer can qualitatively improve the DNN programming skill. We contribute to Human-Computer Interaction (HCI) as follows:

- A literature study on DNN visualization based on figures from machine learning academic papers and existing visualization tools. From it, we summarize visual principles for the DNN programming environment.

- A novel DNN programming environment for teaching non-expert programmers about DNN modeling and programming paradigm. The programmers can edit the neural network in the editable live visualization and check the tensor shape in real-time.

- An exploratory user study that shows Konjak helps in two

aspects: novice machine learning users could touch the DNN programming paradigm and fix layer-tensor alignment during programming; experienced DNN programming trainers could teach the skills to the learners by demonstrating text-based editing and the corresponding effect on the visualization, or vise versa.

## 2 RELATED WORK

### 2.1 Deep Neural Network Bug and Repairing

In practical DL system development, the developers mainly use modern DL frameworks like Tensorflow [1], Chainer [35], Pytorch , [29] and Keras [10] to programmatically build their model. These embedded domain-specific-languages (DSL) provide packaged functions and layers in DNN programming and keep updated to support the newest statistical functions proposed in the machine learning community. Many studies have researched on challenges that a programmer may face in DL system development. Amershi et al. surveyed software engineers from Microsoft teams. They found that the more the programmer experiences machine learning software engineering, the more they consider the use of "AI tools" as a challenge [4]. Cai et al. investigated software engineers' motivation, hurdles, and desires in shifting to machine learning engineering and found that "implementation challenge" is unignorable [7]. Zhang et al. and Islam et al. looked deeper into the DL programming process by analyzing posts from *StackOverflow* and repositories on *GitHub* [19, 39]. According to them, "Program Crash" is a common category of bugs in the deep learning system, and amongst this type, "Shape Inconsistency" is one of the most questioned bug types. "Shape Inconsistency" refers to runtime errors caused by mismatched multi-dimension arrays between operations and layers [39]. To ensure that the shape of the array will not differ from the developer's desired mental model, the programmer may keep calling *print statement* or have their model visualized after the editing towards the network definition program file is over, then repeat the process until they are satisfied with the network's structure. Konjak is designed based on our observation of this specific type of bug. With the "Level 3" liveness in DNN programming (as explained in the following subsection), the novice programmer can more efficiently master the mechanism of DNN development.

### 2.2 Coding-free DNN Development Tools

In response to the growing desire to become a more professional programmer in data science [4,7,24], Konjak plays the role of a novel DL programming learning environment. AutoML [13] provides a commercial online service to allow the user to upload their data and have their data automatically analyzed without touching the complicated machine learning algorithm and program. It is end-to-end, which means the only thing the user needs to do is to prepare desired input and output data pairs, and the algorithm will pick a proper model and train it automatically. Thus the pipeline is simple enough for those end-users who even don't have coding experience at all, but it is also not originally designed with teaching machine learning programming paradigm in mind. *Neural Network Console* [32] by Sony and *DL-IDE* [33] by IBM provides the user with a fully graphical interface for DNN modeling using block-based visual programming language, without exposing codes to the user. From Tanimoto's classification [34], both of them utilize a "Level 2" liveness in DNN modeling, which means the network structure diagram is editable and executable but not always responsive to the user's edits. However, these coding-free DNN development tools are limited in capability and flexibility. You can do much more by direct coding using frameworks. From the study by Qian et al. in [37], in data science model building, experts prefer graphical tools for communication and education, while non-experts prefer code editor where they are able to start from existing codes. To fill the gap by providing both ways in the user interface, Konjak

is initially designed for novice's DL programming education. We retain the code editor and experiment "Level 3" liveness in modern DNN modeling since its visualization and codes update in nearly real-time whenever the user edits the model in its interface.

### 2.3 Live Programming Environment

Our system is categorized into live programming environments, a concept that already has a long history [27, 34]. The main core of these techniques is the liveness in the programming experience of the under-development program, which is demonstrated by continuously providing the programmers simultaneous feedback so that an evaluation phase can be integrated with the code-editing phase to some degree [27]. By reducing the latency between writing the code and checking its behavior [34], the programmer will have a better comprehension of the behavior they are conducting towards the program. A number of live programming environments have been implemented for different domains while with corresponding features emphasized. For example, TextAlive [22] allows editing computer graphics animation algorithms whose rendering results are shown next to the code editor. Omnicode [20] provides an always-on visualization that presents all numerical values throughout the whole execution history in order to provide the user with better program understanding; Projection Boxes [26] are interaction techniques to enable on-the-fly configuration of such always-on visualizations; Sketch-n-Sketch [17] contributes an output-oriented programming interface to bidirectionally (code ⇔ screen) create and manipulate graphical designs (scalable vector graphics [SVG]); Plotshop [5, 6] augments a text-based code editor with an interactive scatter plot editor to enable the user to author synthetic 2D points dataset for machine learning algorithm test more intuitively. Skyline [38] is quite a similar project with Konjak in concept and background but focuses on in-editor DNN computation performance profiling.

## 3 BACKGROUND ON DNNS PROGRAMMING

As stated in Sect. 2, many previous works have focused on obstacles that a user might meet during writing DL programs and utilizing DL in their systems [4, 7, 19, 39]. In this paper, we especially focus on *model structure comprehension* and *tensor shape inconsistency* during DNNs programming. This section briefly introduces the background of DNNs programming practice and common bugs in DL development.

**a) DNN programming using DL libraries:** Writing a program to define a DNN is actually to assemble a series of mathematical functions into a sequence. The finished function sequence is called a *network*, and each mathematical function is identified as a *layer* in the network, which may have weights. In practice, DL libraries are maintained to provide common layers and tools to build up a network, which has much-eased programmers' burden in the network programming phase. Once the network is assembled up, the programmer writes scripts to define the *training* process, where the DNN is created as an instance, also called *model*, then its weights iteratively learn from *batches* of input data. In their heart, DNN models receive data in specific tasks and output predictions. In each *iteration* of the training process, the model makes a prediction towards the input data batch, and based on errors of the prediction, the model's weights are updated to reduce the prediction error. In this process, layers receive and emit data in the form of a multi-dimensional array, e.g., for an image as the input into the network, the tensor is usually in shape (batch size, channel size, height, width). The multi-dimensional array is heuristically called *activation* or *tensor*. After the training is finished, the programmer *evaluates* the performance of the trained model, and iteratively improves it by muting arguments and network structure. Optionally, the programmer may *deploy* the model into an actual system.

**b) Common bugs in DNN programming:** Bugs in DNNs can be

Table 1: Examples of DNN structure visualization design. We categorized them based on (a) what the node represents and (b) how to draw the node.

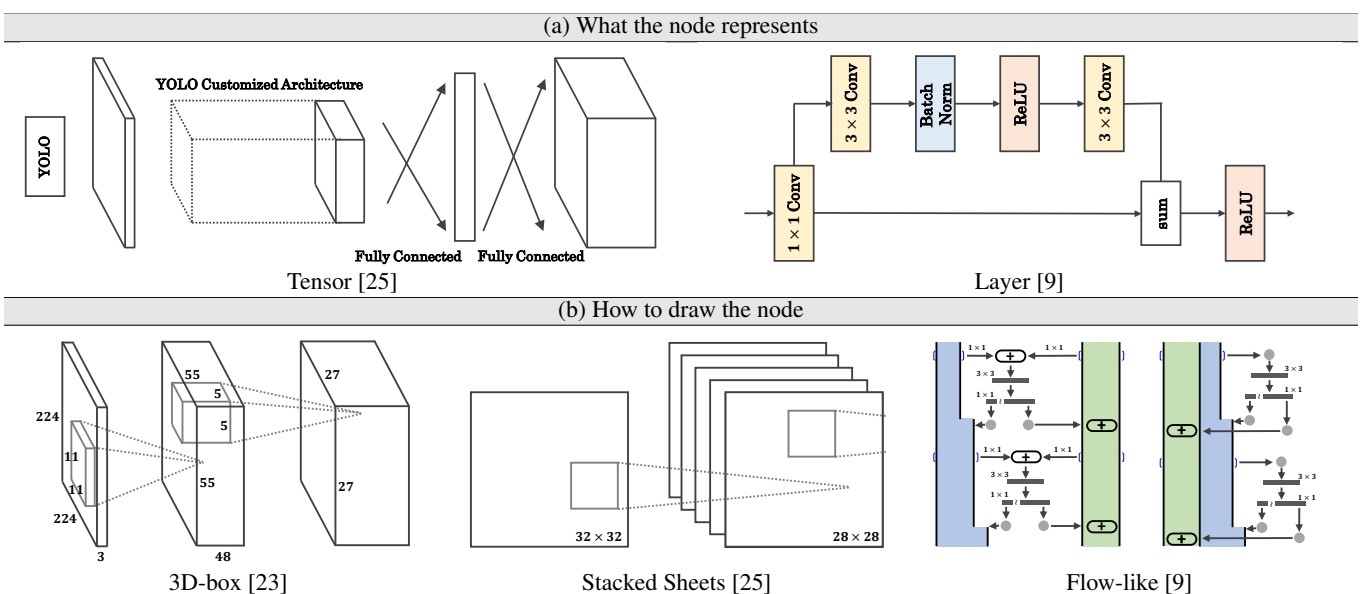

| (a) What the node represents | |
| --- | --- |
| Tensor [25] | Layer [9] |
| (b) How to draw the node | | |
| 3D-box [23] | Stacked Sheets [25] | Flow-like [9] |

identified as explicit or implicit bugs. Explicit bugs are those that crash the program and abort the training or evaluation process. Implicit bugs won't produce any errors during program execution but cause symptoms like abnormal training or low prediction accuracy. In this work, we focus on explicit bugs that crash the program. According to Zhang et al. in their study over DL-related questions collected from StackOverflow [39], the most common explicit bugs that cause program crash in DNN program are: 1) Shape inconsistency: Layers in a network are defined with several arguments and can only receive tensors in a specific shape. If a layer's arguments mismatch with the input tensor shape (e.g., in the image classification task, produces a tensor that has non-integer height or width), the execution will be aborted and throw an error; 2) Numerical error: data in DNNs is mainly defined as float point values [12], and inconsistent numerical type will easily raise an undesired error, e.g., when a float64 tensor is an input to a layer with float32 weights; 3) CPU/GPU incompatibility: GPU plays a vital role in accelerating DNN's training and evaluation iteration but given a trained model and relevant published codes, the execution may fail on another CPU-only machine because the codes are GPU only. These bugs are most frequently asked in DL application-related questions and do not occur in conventional non-DL applications.

## 4 A Literature Study on DNN Visualization

We address the learning of novice in DNN programming by introducing a synchronized visualization bound with text-based programming. Prior to the design of the system, we first investigated common network drawing practices in DNN to reflect the programmer's mental model towards the under-developed model within our interface. We follow procedures to create more effective domain-specific visualization for communication proposed by [3]. We collected hand-drawn DNN structure diagrams from DL papers to build a database. The papers are picked from *Paperswithcode.com* [28] by visiting leaderboards in three computer vision areas (i.e., image classification, object detection, and semantic segmentation). Besides referring to DL papers, we also get insights from the existing DNN model visualizers, which automatically renders a trained model matrix or manually input network definition into a visual representation.

We analyzed the DNN visualization database (including diagrams from papers and synthesized by tools) and categorized the visualiza-

tions in terms of visual encoding. In the DNN visualization database, it is common to draw network structure in a node-link diagram. We noticed that there are two branches in the design decision ***what the node represents*** (see Table 1): some diagrams choose node as the visual factor when representing the tensors, and the adjacent link represents layers to process tensors; Other diagrams, on the contrary, emphasize layers as nodes in visual representation, in this case, links indicate tensor's flow in the network.

Another important choice in visualization design is ***how to draw the node*** in the diagram, especially in diagrams where tensors are emphasized as nodes. In our observation, three answers are given in this choice: One is to draw the node in a 3D-box, which is the style adopted in network structure diagram by AlexNet [23], the paper opened DL's new age in 2012. Tensor data in DNN can be a 1-D vector or an n-D array, and each dimension's size affects the network's correctness. Drawing tensors in 3D-boxes gives the user an instinctive representation of the tensor's concrete shape; The other style is to draw the node in Stacked Sheets. Here tensors are represented as some stacked rectangles. The size of the rectangle is the encoding of a tensor's map sizes (width and height), and the number of rectangles is encoded from the channel number of a tensor. This style is adopted by LeCun et al. in 1998's famous DNN structure LeNet; a Very limited amount of diagrams choose a flow-like style to represent the flow of tensors. This style encodes tensors into some "rivers," and the river's width represents the tensor's channel number. However, the tensor map's width and height information is omitted in this drawing style. On the other hand, in diagrams where a node represents a layer, the node is usually drawn as a plain rectangle.

As stated in the previous section, "Shape Inconsistency" is the main problem we want to solve in a novice's DNN programming learning. Therefore, in the first choice ***what the node represents***, we pick tensors as the majority of the structure graph. In terms of the second choice ***how to draw the node***, we choose 3D-box to show the tensor's shape as much as possible. Nevertheless, the layer's information should be contained in the visualization. Thus in our design, we divide a separate panel from the interface to list information of all the layers defined by users in code.

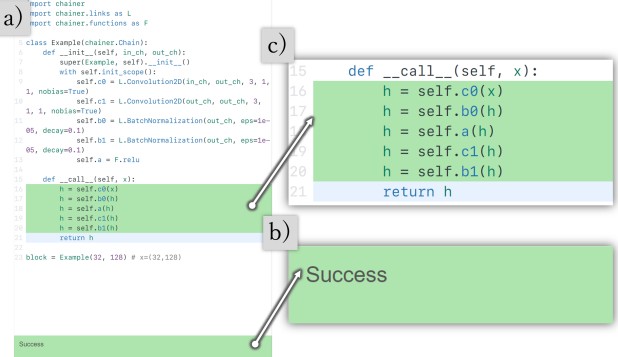

Figure 2: Screenshot of the code editor at left half of the user interface: (a) the code editor, (b) the program error log panel and (c) the inline shape consistency indicator.

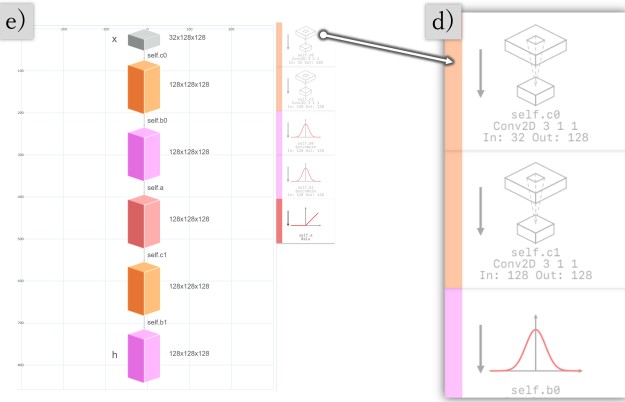

Figure 3: Screenshot of the live visualization at right half of the user interface: (e) layer card bar to list all layers defined in the program, and (d) the interactive graph visualization panel for visualizing network structure and tensor shape.

## 5  KONJAK

We implemented a prototype system, called Konjak, to show the feasibility and effectiveness of live visualization for teaching novices DNN programming and accelerating their progress to DL software engineering. It is implemented as a web-based application written in JavaScript. We use Python as the user-facing language and Chainer [35] as DNN API. We choose Chainer as the DNN library because it was one of the most popular DNN libraries at the time of the implementation; meanwhile, its pioneered dynamic eager execution characteristics has deeply inspired later DNN libraries' API design [2, 29]. Keeping the education purpose in mind, we now visit every component in Konjak's interface and explain their design motivation and functions. The user interface consists of two tightly interlinked main components, the *code editor* and the *live visualization*.

### 5.1  Code Editor

The left half of the screen is split as the code editor component, where the user writes a program like in a standard workflow of general-purpose programming (See Fig. 2). This consists of three child components: a) text-based code editor, b) problem panel, and c) inline shape check and highlight.

**a) Text-based code editor:** We utilize CodeMirror [14] as the backbone of the text-based code editor in a browser. The user writes a DNN structure program in this area. Note that although in the prototype we only support Chainer, without loss of generality, the interface can be transferred to other popular DNN libraries like PyTorch due to their similar API design. A DNN structure in Chainer usually starts from inheriting a built-in class `chainer.Chain`, which is a class to define a neural network composed of several layers (`chainer.links` or `chainer.functions`). In actual code structure, the user defines layers to use in the network in function `__init__` by assigning `Chainer.links` instances to attributes. For example, the code `self.c0 = L.Convolution2D(3, 16, 3, 1, 1)` assigns a 2D convolution layer instance to a attribute `c0`, with the convolution layer's arguments given as "Use a $3 \times 3$ kernel to convolute a 3-channel tensor in stride 1, and output 16-channel tensor. In convolution, the input tensor map is spatially padded with width 1 for each channel".

The other necessary component in code is function `__call__`, where the user gradually connects layers defined in function `__init__` to input data. When the network definition is over, the user is supposed to give a shape definition of input data in a comment, e.g., in line 23, `x=(32, 128)`, means `x` is a tensor in the shape of $32 \times 128 \times 128$. In our current prototype, we assume that

the input data's width and height are the same for simplification. Konjak provides a live programming environment that always parses the under-development program and updates its visualization. Every time the user stops typing after 0.5 seconds, the browser will send the program back to a server to have it checked for further visual feedback.

**b) Problem panel:** After the program is sent to the server, its syntax will be checked first by the library Pylint. If no errors are found, the server parses the received program into AST and sends simplified AST back to the browser in JSON format. Konjak renders the received JSON into visualization in the user interface and prints an error message in the problem panel, which can be syntax errors detected by Pylint or non-syntax errors (tensor shape inconsistency). The state of the program is encoded to panel background color to notify users, where red is for error and green is for success. The line number is included in the message to help users locate errors.

**c) Inline shape check and highlight:** Synchronized with the problem panel, in-situ tensor shape inconsistency indicator is activated in lines where `__call__` function is defined. Applying exactly the same color encoding as the problem panel, the line where the inconsistency happens will become red. The indicator stops showing color at that line. Otherwise, all lines in the `__call__` function show green.

### 5.2  Live Visualization

Live visualization (See Fig. 3) is designed to help novices lively check DNN structure and interactively connect layers to tensors lies in the right half of the user interface. Following the design principles we described in the last section, we use two sub-panels to visualize the neural networks: d) layer card bar and e) graph visualization panel.

**d) Layer card bar:** We retain a bar to list all layers defined in function `__init__`. In this bar, all the layer instances assigned to the network's attributes are drawn in separate cards, and Konjak places the cards in the narrow bar vertically, following the order that layer instances are created in the program. These layer cards are responsive to the program in the code editor. When the user adds or deletes lines in the program to create or delete layer instances, the corresponding layer cards will simultaneously arise or disappear. Taking the characteristics of different types of layers in DNN into consideration, we present a brief and explanatory layer card design.

Fig. 4 shows some examples of layer cards in Konjak. For the

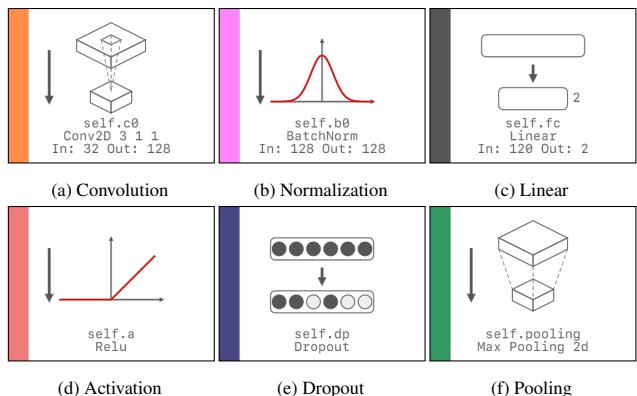

(a) Convolution     (b) Normalization     (c) Linear

(d) Activation     (e) Dropout     (f) Pooling

Figure 4: Examples of layer card designs.

top 2/3 space of the card, we draw different sketches to represent different types of neural network layers. These brief sketches also indicate the layer's effect on the tensor shape, e.g., a linear layer will flatten a tensor into a fixed-length vector, and a max-pooling layer down-sample a tensor by picking the maximum value in each tile of the tensor to reduce original data volume to a smaller one. We show the layer's parameter information at the bottom 1/3 space for those layers that only receive and emit specific data size. At the left end of the cards, we encode different layer categories into a colored strip (Orange for convolution, purple for normalization, ash for linear, red for activation, dark blue for dropout, and green for pooling). Every time the user edits parameters to create a layer instance in the code editor, the corresponding layer card updates its drawing as well as information.

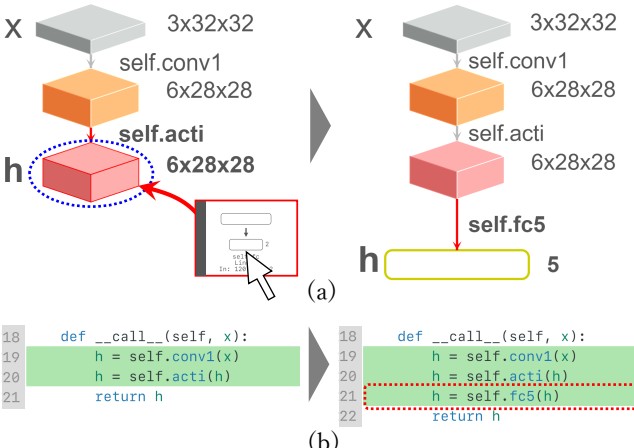

Figure 5: Drag-and-drop function in Konjak's live visualization. (a) the user is allowed to drag a layer card and drop it on the tensor node they want to connect the layer to, then the graph visualization updates to reflect the connection, meanwhile (b) new code line (red frame) is synthesized simultaneously in code editor.

**e) Graph visualization panel:** this is the heart of Konjak, which interacts with all other components in the user interface. Similar to the layer card bar, it mainly provides the user synchronized visual representation of the network structure in the form of a node-link graph. To keep the user's image towards tensor shape clear, we draw 3D tensor nodes in 3D-box and 2D tensor nodes in the 2D bar, with tensor shape printed next to the box ($Channel \times Height \times Width$). Color encoding of the cube or the 2D bar is consistent with the

layer card but to indicate layer type that reproduces the tensor. As a special type of tensor, we encode input data as grey cubes. $h$ or $x$ at the side of tensor nodes means these tensors are assigned to variables in the program.

Graph visualization panel interacts with the layer card bar by providing a drag-and-drop feature. The user is allowed to click and drag the layer card, then drop it on these tensor nodes with the variable names to connect the layer to the tensor node. After the drag-and-drop operation, a new node will show as the next tensor, and variable information updates simultaneously. This operation will also affect the code editor as a way to synthesize new code lines in relevant line numbers from visualization. We implement this feature by binding the visualization elements with the relevant line number and offset information in the program. When the drag-and-drop operation is conducted, Konjak attaches the synthesized codes after the relevant last line, then the browser sends the updated program back to the server and re-render the visualization based on the response from the server. The editable visualization, together with the synchronized update from code editing to visualization, builds a bidirectional DNN editing experience for novices. To make the binding between program and visualization tighter, when the user move cursor in the code editor, the line cursor movement will trigger highlights in relevant visual parts. In reverse, if the user's mouse hovers on link (layer) or node (tensor) in the graph visualization panel, the corresponding layer card will show a different background color, and relevant code lines will become bold.

## 6 USAGE SCENARIOS

Under the context of DNN programming curriculum and a novice's first DNN programming exploration, we present two usage scenarios here as examples to show how Konjak can be used.

### 6.1 Convolution layer setting playground

Compared to other layers that don't affect the input tensor's shape or simply reduce the original tensor's size to half, the convolution layer's output tensor shape is influenced by many layer parameters in the definition. The formula to calculate the output tensor shape of a convolution layer is:

$$C_o = K$$
$$W_o = \frac{(W_i - F + 2P)}{S} + 1 \tag{1}$$

where input tensor is of size $(C_i, H_i, W_i)$, output tensor is of size $(C_o, H_o, W_o)$, and the convolution layer's arguments are defined as `L.Convolution2D(in_channels=W`$_i$`, out_channels=K, ksize=F, stride=S, pad=P)`. Some specific parameter settings are commonly used in neural networks implementations, like (`ksize=3, stride=1, pad=1`) to keep input tensor size and (`ksize=4, stride=2, pad=1`) to reduce the size

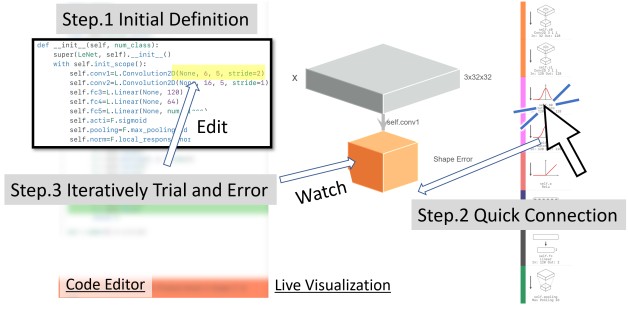

Figure 6: Konjak enables DL programming learners solving shape consistency problem iteratively in much lower trail-and-error cost.

to half. Konjak can act as a playground for a novice to easily experiment with different convolution layer parameter settings, always with instant visual feedback to help them what is happening with the parameter.

## 6.2 Solving shape inconsistency

Live visualization largely reduces novices' trial and error cost in solving the bugs related to a tensor's shape. Let's consider a scene where a novice is re-implementing a network structure to fit their own application needs. In Step.1, the user starts by copying and pasting a template of a DNN program, then defines the layers that will be used in the network. Layer parameters don't need to be precisely filled at the beginning because Konjak's visualization is synchronized. Even when there is improper parameter-picking, our system can tell the user where it is wrong. With layers defined with randomly picked initial parameters in the code editor, the user can use the drag-and-drop function to have a quick network connection in Step.2. Then, the user will notice the error message shown in the problem panel and inline highlight and locates the layer that results in the shape error. Step.3, the user is allowed to check the live visualization alternately, go back to the layer definition part and modify the parameters, and repeat this process until the tensor shape shown in the visualization is satisfactory. Compared to current practice to use a print statement or visualization tool after the code editing phase, trial and error cost here is reduced to less than one second.

## 7 USER STUDY

As an educational tool to help novices learn DNN programming, we hope Konjak's liveness and visualization can 1) help form a proper mental model towards DNN structure and 2) shorten the cycle in input/feedback in the learning stage. To evaluate the points above in the educational context, we ran an exploratory first-use study to get feedback from both DL novices (learners) and experienced DL developers (trainers). We invited 12 participants (nine males and three females), aged 22 to 28, with programming experience from two to 10 years. P1 - P4 are Deep Learning engineers who mainly act as the role of a trainer, and P5 - P12 are lab internal members who are identified as novices in DNN programming (has nearly no knowledge or only basic knowledge about DNN). Note that we determined the number of participants (sample size) based on the professional standards for this type of study within the HCI community [8].

## 7.1 Procedures

The study lasted 60 minutes for each participant. We started with a 15-minutes warm-up session on Konjak's user interface and features, as well as the basics of DNN programming. During the session, the participant was allowed actually to touch Konjak to get familiar with the system. Next, we presented two tasks that are common in the DL programming learning situation and have been mentioned in Sect. 6. We designed the tasks by talking to expert machine learning users. Each task lasts 15 minutes. After the task phase, we conducted an interview and questionnaire in the 15 minutes remaining. Note that all the participants worked on the same tasks. P5 - P12 are the main target participants to simulate the DNN programming learning situation, and P1 - P4 are mainly recruited to provide extra reviews about our system from the perspective of an educator. The tasks are:

- **Task 1**: Given desired input and output tensor size, explore proper layer parameter settings to finish the tensor shape transformation.

- **Task 2**: Given the DNN structure diagram cropped from a research paper (LeNet [25]), try to implement it in Konjak.

Table 2: Summary of post-study questionnaire results.

| # | Items in Questionnaire | $\mu$ | $\sigma$ |
|---|---|---|---|
| 1 | *The live visualization diagram reflected my mental model towards the network structure well.* | 4.42 | 0.51 |
| 2 | *Highlight features (code2visualization, visualization2code) helped me to locate code/visual element from each other.* | 4.00 | 1.04 |
| 3 | *The design to divide visualization panel into graph visualization panel and layer card bar is intuitive for Chainer's code structure.* | 4.67 | 0.49 |
| 4 | *Using node to represent tensors in visualization, and the shape check feature, helped me in exploring a shape-consistent solution in DNN implementation.* | 4.58 | 0.67 |
| 5 | *Konjak's network and layer card drawing are confusing for me to understand.* | 1.92 | 0.90 |
| 6 | *The consistency between code editor and visualization, helped me observing DNN program more conveniently.* | 4.50 | 0.67 |

## 7.2 Result

All except two (P8 and P11) of our participants finished the two simulation learning tasks using Konjak within the given time. Table 2 shows post-study questionnaire results, which are summarized from six 5-point Likert scale questions (From Strongly Disagree 1, to Strongly Agree 5). These questions covered the core concept of utilizing live programming in the DNN educational context and detailed design points of the visualization. In terms of the drawing of sketches in layer card and network graph to show different layer's functionality, because they are mainly originated from our understanding, we included a question in the questionnaire to survey users' reaction towards our concern that the drawing may cause novices' confusion. We also surveyed participants willing to use live visualization to train other novices in DNN programming in the future. Here we summarize their feedback in three aspects:

**Assist DNN implementation and debug:** some feedback can be categorized into description about how the live visualization help some of our participants implement DNN program and debug parameters. Compared to traditional practice to check tensor shapes using built-in print statements, Konjak enables a shorter route in debugging the DNN program. P6 said: "With the highlight feature and real-time visualization, I can write the program while making sure whether previous lines are correct or not." Exactly as why we are motivated to design Konjak, P7 expressed that the system indeed reduces shape inconsistency in DNN implementation: *"Konjak's interactive feedback and tensor shape inconsistency error messages were very useful to create a DNN structure without worrying about tensor shape inconsistency too much."*

We recorded how the participants used Konjak in the study. When reviewing the video, we noticed P10's whispering in Task 2: *"The input image is (3, 32, 32), and the first feature map on the diagram is (6, 28, 28), so I should first fill in `out_channels=6`... Then what about the parameter `ksize` and `stride`? It seems that `stride=2` will make the output map size decrease too much, so I keep it 1 first... And `ksize=1`, this will make the width 32... (Change the value to 4) Oops, it (map size) is still not small enough... (Re-input 5) Okay, now it's the same as the diagram. So the next layer is a max-pooling layer..."* Observing his behavior in Konjak, we found that it exactly matches the usage scenario of solving the shape inconsistency we presented in the last section.

A skilled DNN programmer also provided positive feedback about the live visualization in aiding DNN implementation. P1 presented a view about the similarity between Konjak's programming experience

and web development: *"This reminds me of my web developing experience, where I put a code editor on the left half and browser on the right half. Since I use hot-reloader, which automatically refreshes the browser whenever I change the code base, I have the tendency to focus on coding and use the visualizer only for reference."* In this sense, we may potentially extend Konjak's features for more than an educational purpose.

**Aid DNN programming education:** Eight participants (P5 - P12) are novices or beginners in DNN programming, and Konjak is highly regarded by them. P8, a CS student who had learned DL before, but nearly forgot all of the knowledge, appreciate the live visualization a lot: *"Although I almost forgot DNN, this structure helped me understand how the layers and structures work. I want to use it when I use DNN in the future."*. P10, identified as a DL learner, said: *"(Konjak) is especially helpful in implementing a DNN while referring to a research paper."*

All of our participants agreed that if it is possible, they tend to use a live visualization like Konjak to teach other beginners DNN programming in the future. From the perspective of a skilled DL engineer, P1 described the situation where he uses a system like Konjak to teach a novice DNN programming: *"If I am going to teach someone Chainer, I probably will use this UI because after understanding how it works, the synchronization between code editor and graph is very helpful to teach the student. I will focus on teaching the student how to write code on the left panel, but occasionally if they make any mistake or don't understand what is going on with the code, then I will remind them to play with the visualization to understand what is going wrong so he can clarify his question and return to coding."*

We ran this study like a get-started lesson to our novice participants, and for those trainer participants, it might seem like a chance for them to think about how to teach the DNN programming paradigm to a newbie. Nevertheless, the programming paradigm is only a side in DL programming. According to the study by Cai et al. [7], other obstacles like mathematics knowledge are still prohibiting novices' diving in.

**Implementation Issues:** Some feedback are about some interface details that affected participants' programming experience. P5 complained about the switching between the code editor and the live visualization: *"It bothered me a little if I have to watch the visualization but edit the network on another side. I'd like to modify the network parts right in the visualization panel."*. P9 suggested another style to draw 3D-Box in graph visualization panel: "Live visualization to show DNN structure is easy to understand. But the drawing of tensor nodes confused me at first. Maybe you can draw it like stacked slices, which can be a more intuitive representation of channels."

## 8 LIMITATION AND FUTURE WORK

Konjak's current prototype is originated from the motivation to support novices in the learning stage. For this reason, only limited layers and structures are supported to be displayed in the live visualization. We implement the prototype to only support convolutional neural networks (CNNs) for tasks like image classification, object detection, and image segmentation, while other structures such as recurrent neural networks (RNNs) and generative adversarial networks (GANs) are out of scope. Fig. 4 covers nearly all the supported layers in the current prototype, nine layers in total. Besides the layers drawn in the figure, two activation layers (`Sigmoid` and `Softmax`) and one normalization layer (`Local_response_normalization`) are supported in the implementation. These six layer categories are typical because they are the main components of some classical DNNs, such as VGG16/19 [30], AlexNet [23], and ResNet [16]. In each category, we only implemented at most three layers because the layers classified into the same category share similar APIs and

visual representation. In DNN programming learning, the DNN model that novices explored might be relatively simple, but when the programmers become skilled, more complicated networks (e.g., with skip link or multiple tensor flows) and customized layers are common in practical development. In our user study, the skilled machine learning programmer participants agreed that the live visualization to always show tensor shape could be quite helpful even in their daily DNN programming. Therefore, the direction to extend Konjak's concept to more situations is worth considering. Features like sub-nets or customized layer visualization are in our future plans. Konjak's current implementation is based on static analysis, which limits the system's application in an expert's practical development. We think that hacking a Python interpreter and tracing its execution memory by executing the actual code will greatly improve the system's extensibility and make it ready for practical development.

In our user study to evaluate Konjak as an educational tool, we design the study like a lesson to train novices in solving shape inconsistency problems and collect their first-use impression of Konjak. We didn't compare Konjak with the print statement or other coding-free DNN network modelers. As far as we know, Konjak is the first research to explore a live programming environment in text-based DNN programming. We hope this work can become the first step toward the acceleration of research in improving DL developing experience in the future.

Also, our participants identified some implementation issues in Konjak's prototype. We thought that they could be overcome with engineering efforts.

## 9 CONCLUSION

We have proposed a system called Konjak that augments a text-based code editor with a synchronized, editable, and representative live visualization to support novices in DNN programming as an educational tool. We revisited DNN structure diagram designs from research papers in machine learning and existing DNN visualizer and extracted design principles, especially for educational context and live programming environment. The system provides a bidirectional editing manner between the code editor and the live visualization and instant tensor shape checking features to avoid the common shape inconsistency error of the DNN program. An exploratory user study was conducted to evaluate Konjak in an educational situation.

### ACKNOWLEDGEMENTS

Removed for review.

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
