# OpenReview forum: "Konjak: Live Visualization in Deep Neural Network Programming as a Learning Tool"
_graphicsinterface.org/Graphics_Interface/2021/Conference — Submitted to GI 2021_

### Official Review · AnonReviewer3 · 2021-01-05
**GI 2021 submission 4**

**Rating:** 6
**Confidence:** 2

**Review:**

This paper proposes a novel user interface for coding deep neural networks. This tool allows to visualize your network both as the executable code as well as a stack of boxes.

I am not an expert in HCI and I do not use visualization tools for my DNN designs very often. However, it seems to me that the proposed tool is sound. The user study also suggests that it could address common problems encountered by novice users and has value for educational purposes.

I think the weakness is that it can mostly help with relatively simple errors, such as shape mis-match for tensors. There is definitely more value in a "training-time" or "inference-time" tool that enables to take a deeper look at neural responses in different layers to gain a deeper understanding of why network gives a particular output for a specific input. I think that mismatches in tensor shapes, while common, are fairly easy to debug once you gain some experience with NN modules. However, debugging an undesired behavior is much more challenging and under-explored.  I also wonder, how this tool would extend in more complex architectures (e.g., Siamese architectures or the ones that use a more complex tensor graphs).

---

### Official Review · AnonReviewer2 · 2021-01-14
**Interesting direction, but lack of rigor throughout.**

**Rating:** 3
**Confidence:** 4

**Review:**

This paper presents a system, Konjak, that synchronizes code editor and a visualization of the resulting Deep Neural Network structure. The purpose of these synchronized views is educational, i.e. targeted at helping DNN non-experts program DNN. The system is evaluated through a light user study.

The technical aspects of the paper are well presented and relatively easy to follow. The process (formative study, design, usage scenario, evaluation) is convincing. What is less convincing, however, is the execution of this plan. As I detail below, the formative study does not follow any particular methodology; the target users (novices) were not consulted or involved in the process at all until the final evaluation; the design decisions are not well justified; and the study is not convincing in its rigor, execution and interpretation.

Not being an expert in DNN myself, I appreciate how the related work clearly explains how existing tools and approaches relate to Konjak. I would expect, however, that more explanation be given regarding how Skyline differs from Konjak. The authors state that both tools share concept and background, so I would expect more details about their differences to clearly identify how the proposed contribution differs.

The literature study is interesting in its aim; however its execution is not convincing. I am missing a methodology section that would explain how this study was actually conducted. There are dozens of ways of conducting such reviews and many would be valid approaches, but the lack of any methodology raises questions about soundness and rigor of the study, seriously limiting the usefulness of the results. Aspects that are missing include information about who conducted the study, how many people were involved, how many diagrams were studied (and I would also expect this dataset of diagrams to be accessible though supplemental material), which analysis/coding method(s) were used, how themes were identified, etc..

One key concern I have with the paper is that there are many claims made about how the tool is designed for novices, however, none of the formative work actually relates to novices. From a methodological perspective, does it make sense to review papers, likely written by experts on the topic? Would not the results tell us how experts represent these DNN? Why not asking novices to draw DNNs to extract the mental model that novices might have? This would lead to more insightful findings with regards to designing a tool for novices.

I had a hard time making the connection between the study and the live visualization. Although the authors write that (paraphrase) **the design process follows the design principles described in the last section**, it is unclear to me what these design principles are, because they are not clearly articulated. It is left to the reader to attempt to make these connections, that should be more explicit. Right now, the design decisions seem quite arbitrary (e.g., why layer cards? why these ones?)

Similar to other parts of the paper, the study, although generally adapted to its purpose (qualitative and exploratory) has too many aspects that lack grounding/justification. First, if it is a tool for novices to learn about DNN, why involving experts? There might be some good reasons for that, but they are not presented (why is it useful to have this educator perspective?). Also, "We designed the tasks by talking to expert machine learning users" is not a well-rounded explanation for why these tasks were selected, how they were formulated, which alternatives were considered, what coverage they have, etc..
One particularly concerning aspect of the study is that it is half-baked quantitative and half-baked qualitative. The number of participants is very reasonable for a qualitative study that could give some rich insights into how novices use the tool. However, this would also require following a rigorous methodology and go in much more depth than what is presented in the paper. The alternative is to be more on the quantitative side, but in this case the sample size should also be larger and less subjective data should also be looked at. There are also issues with the questionnaire, for example asking people if a representation accurately reflects their mental model of something after having them learn and use this mental model is biased.
Overall, the study is too superficial and any given representation of a DNN would have led to positive results (of course having some kind of representation rather than just code will help). This does not mean that the designs implemented in Konjak are good.

---

### Official Review · AnonReviewer1 · 2021-01-15
**Interesting but flawed study**

**Rating:** 5
**Confidence:** 4

**Review:**

The authors of this paper describe the design and prototype of a system to visually specify and identify the shape of tensors in a deep neural network (DNN) intended to highlight and help the user correct shape inconsistencies. This is a common bug and difficult to tell in real-time (where real-time refers to during the programming specification of the DL network.).  In the system Konjak the user creates the tensors and connects them using drag and drop interaction in an incremental specification.  The system then generates the appropriate code.   The premise is that real-time interactive visual specification both prevents errors and in doing so highlights how the networks operate  and thus scaffolds learning.
The authors carried out a user study to assess the utility and viability of the method.

I am not expert in the educational challenges of DNN comprehension, but I very much liked the motivation of this paper. Unfortunately the work as reported suffers flaws that discourage me from recommending acceptance.
The authors do a very poor job of comparing this approach to other DNN visualization and/or bug correction techniques.  The most well-known DNN visualization and specification tool , often used to scaffold novices, is the TensorFlow playground.  Programmatic tools like VariFlow provide code-level debugging detection and correction capabilities. How does konjak fit into and compare to these tools and use cases? What does it bring?

The user study is a classic example of a poorly designed study. If this study was supposed to examine how much better (or worse) that Konjak is than traditional coding tools, then why not study that? Why not do a between subjects study where one group tried to achieve the tasks with standard DNN   programming methods and the second group used Konjak? AS it si the study is a thinly vweiled attempt to confirm the assumption that this is valuable. MOreover, if the purpose of the study is to assess the utility for novices, why do experts form 1/3 of the participants?  The standard a=way to approach this is typically an expert review as parallel or separate sessions, and to run a reasonable number (where 12 is a minimum for a repeated measures study, and quite small for a feasibility study).   The authors make assertions like "participants tend to use a live visualization" as opposed to stating that participants expressed an opinion that they would use it if it were available. Very different. This is an example of poor reporting and not results misinterpretation, but it does not stand up to the kinds of scrutiny we would expect in user study results.

---

### Meta-Review · Area_Chair1 · 2021-01-15

**Recommendation:** Reject
**Confidence:** 4

**Metareview:**

Reviewers appreciate the motivation for the work (R1, R3) and the aims of the research (R2).

Although R3 writes that the tool seems sound, both R1 and R2 argue that the paper is not ready for publication given its too many (important) issues. These key issues are as follows:
- lack of comparison with existing related tools and highlight of key differences/contribution (R1, R2)
- poorly designed and executed study (R1, R2)
- misalignment between purpose of the tool and study aims (R1, R2)
- lack of methodological groundings throughout (R2)

R3 is more positive about the work, while also providing some comments regarding limitations of the tool that should be discussed in the paper.

Overall, the issues raised by reviewers are too important/major to be addresses in a short revision cycle. Given that all reviewers found the direction promising and valuable, I strongly encourage the authors to consider the detailed reviews and suggestions for improvement toward a future submission.

---

### Decision · Program_Chairs · 2021-01-16

Reject